# Predictors of functional improvement in the short term after MitraClip implantation in patients with secondary mitral regurgitation

Michael G. Paulus[1]*, Christine Meindl[1], Lukas Böhm[1], Magdalena Holzapfel[1], Michael Hamerle[1], Christian Schach[1], Lars S. Maier[1], Kurt Debl[1], Bernhard Unsöld[1], Christoph Birner[2]

1 Department of Internal Medicine II, University Hospital Regensburg, Regensburg, Germany, 2 Department of Internal Medicine I, Klinikum St. Marien, Amberg, Germany

☯ These authors contributed equally to this work.
* michael.paulus@ukr.de

## Abstract

### Background and objectives

MitraClip implantation is an established therapy for secondary mitral regurgitation (MR) in high-risk patients and has shown to improve several important outcome parameters such as functional capacity. Patient selection is both challenging and crucial for achieving therapeutic success. This study investigated baseline predictors of functional improvement as it was quantified by the six-minute walk distance (6MWD) after transcatheter mitral valve repair.

### Methods and results

We retrospectively analyzed 79 patients with secondary MR treated with MitraClip implantation at an academic tertiary care center. Before and four weeks after the procedure, all patients underwent comprehensive clinical assessment, six-minute walk tests and echocardiography. 6MWD significantly improved after MitraClip therapy (295 m vs. 265 m, $p < 0.001$). A linear regression model including seven clinical baseline variables significantly predicted the change in 6MWD ($p = 0.002$, $R^2 = 0.387$). Female gender, diabetes mellitus and arterial hypertension were found to be significant negative predictors of 6MWD improvement. At baseline, female patients had significant higher left ventricular ejection fraction (49% vs. 42%, $p = 0.019$) and lower 6MWD (240 m vs. 288 m, $p = 0.034$) than male patients.

### Conclusion

MitraClip implantation in secondary MR significantly improves functional capacity in high-risk patients even in the short term of four weeks after the procedure. Female gender, diabetes mellitus and arterial hypertension are baseline predictors of a less favourable functional outcome. While further validation in a larger cohort is recommended, these parameters may improve patient selection for MitraClip therapy.

**Data Availability Statement:** All relevant data are within the paper and its Supporting Information files.

**Funding:** The authors received no specific funding for this work.

**Competing interests:** The authors have declared that no competing interests exist.

## Introduction

Secondary mitral regurgitation (MR) is a very common valvular heart disease and associated with poor prognosis in patients with heart failure [1–3]. Surgical therapy consisting of valve repair or replacement did not show to improve prognosis and is often prohibited by an unacceptable high perioperative risk in patients with secondary MR [4–6]. As an alternative to otherwise conservative management, the MitraClip procedure is an established method for the percutaneous edge-to-edge repair of the mitral valve with a superior safety profile in high-risk patients [7,8]. As two randomized controlled studies recently delivered differing results, the procedure's efficacy in reducing mortality in patients with secondary MR remains controversial [9,10]. Yet, several registry studies consistently demonstrated an improvement in symptoms and quality of life after treatment with the MitraClip procedure [7,8,11], which seems to be a preferential outcome measure in those elderly and multimorbid patients.

The underlying cardiac pathology and mechanism leading to the development of secondary MR are highly variable. They comprise ischemic heart disease, nonischemic cardiomyopathy, annular dilation and abnormal leaflet tethering [12]. Additionally, patients with secondary MR often exhibit significant cardiac and noncardiac comorbidities, which further contribute to the heterogeneity of this collective [4]. As a consequence, selecting patients who will benefit from MitraClip implantation is both important and challenging [13]. Various studies attempted to identify predictors of therapeutic success after MitraClip implantation, mainly focusing on mortality [14–18] and thereby neglecting functional improvement as an equivalent outcome parameter. Therefore, besides improving prognosis, a main therapeutic goal is the reduction of symptoms and improvement in functional capacity. To assess the latter, the six-minute walk test is a widely used tool in cardiovascular research which correlates with echocardiographic signs of MR reduction after transcatheter mitral valve repair [19,20]. In order to improve patient selection, the aim of this study was to identify clinical baseline predictors of the improvement in the six-minute walk distance (6MWD) after MitraClip implantation.

## Methods

### Study population

Patients who underwent transcatheter mitral valve repair by MitraClip implantation at the University Hospital Regensburg from 2011-2019 were analyzed retrospectively. Qualifying inclusion criterion was symptomatic moderate-to-severe or severe secondary MR with or without left ventricular (LV) systolic dysfunction. Indication for MitraClip therapy was given by an interdisciplinary Heart Team consisting of interventional cardiologists, cardiac surgeons and anesthesiologists. The procedure was performed as described elsewhere [21] under general anesthesia, guidance by fluoroscopy and three-dimensional transesophageal echocardiography. Exclusion criteria were intraprocedural failure to implant a clip, conversion to surgery or repeat MitraClip procedure. Also, patients who did not complete the follow-up and/or did not perform a six-minute walk test were excluded in the intent of a complete case analysis. The study was approved by the local ethics committee. As only pre-existing data was analyzed retrospectively and anonymously, consent was not required. The inclusion process is illustrated in Fig 1.

### Clinical and echocardiographic assessment

As part of the routine care at our institution, all patients underwent a clinical and echocardiographic assessment at baseline and four weeks after MitraClip implantation. Evaluation incorporated past medical history, physical examination, laboratory measurements, transthoracic

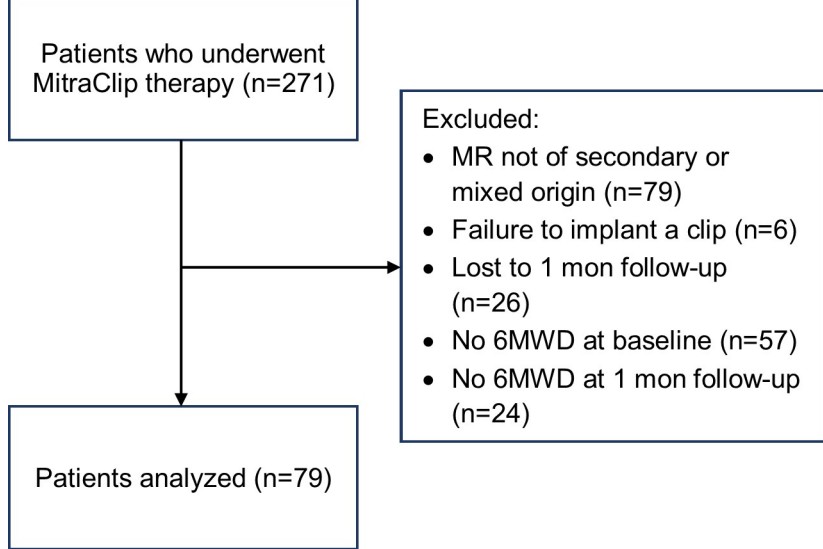

**Fig 1. Flow chart depicting the inclusion process.** Only patients who completed 6MWD at baseline and at the four weeks follow-up were included in the intent of a complete case analysis. mon, month; MR, mitral regurgitation; 6MWD, six-minute walk test.

echocardiography, New York Heart Association (NYHA) functional class and measurement of 6MWD. Additionally, NYHA functional class was also evaluated at a short visit twelve months after the procedure. Echocardiography included measurement of left ventricular dimensions, left ventricular systolic function and quantification of MR in accordance to current guidelines [22]. MR grading was based on color and continuous wave Doppler evaluation including vena contracta width, effective regurgitant orifice area and regurgitation volume estimated by proximal isovelocity surface area method, and regurgitant jet area. MR grade was scored from 1 to 4 (1: mild, 2: mild-to-moderate, 3: moderate-to-severe, 4: severe). Device success was defined as residual MR grade $\leq 2$ after MitraClip implantation.

Six-minute walk tests were conducted as described elsewhere [19]. Δ6MWD was calculated as 6MWD at four weeks after MitraClip procedure minus 6MWD at baseline.

## Statistical analysis

Continuous variables with normal distribution were reported as mean ± standard deviation, continuous variables with skewed distribution as median [interquartile range]. Categorical variables were reported as numbers and percentages. Differences between continuous variables in paired data were tested with a paired t-test, continuous variables in unpaired data were compared with an unpaired t-test. Ordinal variables in paired samples and ordinal variables in unpaired samples were compared using Wilcoxon signed-rank tests and Mann-Whitney U tests, respectively. Comparisons of nominal variables were performed by Pearson's chi-squared tests, comparisons of binary variables by McNemar's test. Correlation between variables was analyzed by calculating Spearman's rank correlation coefficient.

To identify independent predictors of improvement in 6MWD after MitraClip implantation, multiple linear regression was used. Selection of predictors was based on clinical considerations and exploratory statistical analysis. To avoid overfitting the regression model, clinical and echocardiographic predictors were investigated in two separate regression models. Each model included age, MR grade and baseline 6MWD as covariates. Outliers were identified by

calculating studentized residuals and Cook's distance. A two-sided p-value < 0.05 was considered statistically significant. All statistical analyses were performed using SPSS Statistics 25.0 (IBM, Armonk, NY USA).

## Results

### Patient characteristics

In total, 79 consecutive patients who underwent MitraClip implantation between November 2011 and January 2019 were included in the study. Baseline characteristics are shown in Table 1. Mean age was 76 ± 7 years, gender distribution was almost equal (46.8% female).

**Table 1. Baseline characteristics of the study population (n=79).**

| | |
|---|---|
| Age, y | 76 ± 7 |
| BMI, kg/m$^2$ | 26.1 ± 4.4 |
| Female gender | 37 (46.8) |
| Heart failure entity | |
| HFpEF | 32 (40.5) |
| HFmrEF | 19 (24.1) |
| HFrEF | 28 (35.4) |
| Coronary artery disease | 47 (59.5) |
| Dilated cardiomyopathy | 12 (15.2) |
| Prior PCI | 38 (48.1) |
| Prior CABG | 19 (24.1) |
| Prior myocardial infarction | 25 (31.6) |
| Atrial fibrillation | 51 (64.6) |
| Arterial hypertension | 52 (65.8) |
| Diabetes mellitus | 25 (31.6) |
| Chronic kidney disease | 51 (64.6) |
| COPD | 9 (11.4) |
| CRT | 9 (11.4) |
| ICD | 22 (27.8) |
| Logistic EuroSCORE, % | 19.5 ± 12.3 |
| EuroSCORE II, % | 7.6 ± 6.5 |
| NTproBNP, pg/ml | 3618 [1949–5983] |
| Serum creatinine, mg/dl | 1.48 ± 0.63 |
| GFR, ml/min | 47 ± 19 |
| MR etiology | |
| secondary | 73 (92.4) |
| mixed | 6 (7.6) |
| No. of clips implanted | |
| 1 | 50 (63.3) |
| 2 | 29 (36.7) |

Continuous variables with normal distribution are expressed as mean ± SD, continuous variables with skewed distribution as median [IQR]. Categorical variables are expressed as n (%).

BMI, body mass index; CABG, coronary artery bypass graft; COPD, chronic obstructive pulmonary disease; CRT, cardiac resynchronization therapy; GFR, glomerular filtration rate; HFmrEF, heart failure with mid-range ejection fraction; HFpEF, heart failure with preserved ejection fraction; HFrEF, heart failure with reduced ejection fraction; ICD, implantable cardioverter-defibrillator; IQR, interquartile range; MR, mitral regurgitation; PCI, percutaneous coronary intervention; SD, standard deviation.

Preserved left ventricular ejection fraction (LVEF >50%) was present in 40.5% of patients, while 35.4% had reduced LVEF (<40%) and the remaining 24.1% being classified as mid-range LVEF (40-50%). Concerning underlying heart disease, 59.5% of patients suffered from coronary artery disease and 15.2% presented with dilated cardiomyopathy. Atrial fibrillation, arterial hypertension and chronic kidney disease were very common comorbidities (64.6%, 65.8% and 64.6%, respectively). Mean logistic EUROScore and EUROScore II were 19.6 ± 12.3% and 7.6 ± 6.5%, reflecting high perioperative risk. One MitraClip was implanted in 63.3% of the patients, the remaining 36.7% received two MitraClips. Intake of heart failure medication and loop diuretics was highly prevalent at baseline and did not significantly change four weeks after the procedure (see Table 2).

## Echocardiographic and functional data at baseline and four weeks after MitraClip implantation

All patients showed moderate-to-severe or severe MR at baseline (grade 3 26.6%, grade 4 73.4%), which was significantly reduced four weeks after MitraClip implantation (grade 1 70.9%, grade 2 24.1%, grade 3 5.1%, p < 0.001). Thus, device success was achieved in 94.9% of patients. Mean LVEF was 45 ± 14% and remained unchanged in the short follow up after the procedure. Left ventricular end diastolic (LVEDD) and end systolic diameter (LVESD) decreased slightly yet significantly four weeks after MitraClip therapy (59 ± 9 mm vs. 57 ± 9 mm, p = 0.035 and 47 ± 11 mm vs. 46 ± 11 mm, p = 0.049). Data are reported in detail in Table 2.

At baseline, the study population was highly symptomatic, with 77.2% presenting with NYHA functional class III and 17.7% with class IV. Symptoms significantly improved four weeks after MitraClip procedure, when 62.0% of the patients were judged NYHA class I or II (p < 0.001). Improvement in NYHA class remained stable after twelve months (I: 20.0%, II: 52.7%, III: 27.3%, IV: 0%, p < 0.001 vs. baseline). Correspondingly, 6MWD was markedly reduced at baseline and significantly improved at the four weeks follow up (265 ± 103 m vs. 295 ± 104, p < 0.001). Furthermore, postprocedural 6MWD correlated with NYHA class both four weeks and twelve months after MitraClip therapy (r=-0.38 and r=-0.36, see Fig 2). Mean Δ6MWD was 30 ± 68 m and did not differ between patients with baseline MR grade 3 and grade 4 (17 ± 88 m vs. 34 ± 60 m, p = 0.312).

## Baseline predictors of improvement in 6MWD four weeks after MitraClip implantation

A multiple linear regression including eight clinical variables, adjusted for age, MR grade, baseline LVEDD and 6MWD, significantly predicted Δ6MWD from baseline to the four weeks follow up (p = 0.002, see Table 3). $R^2$ of the overall model was 0.387, indicating a high good-ness-of-fit. While no positive baseline predictors of Δ6MWD were found, several independent negative predictors were identified. The strongest negative predictor was diabetes mellitus (B = -46.9, p = 0.002), followed by arterial hypertension (B = -39.2, p = 0.010). Furthermore, 6MWD at baseline and female gender also negatively predicted Δ6MWD (B = -0.2, p = 0.044 and B = -32.3, p = 0.042). Coronary artery disease, dilated cardiomyopathy and atrial fibrilla-tion did not independently predict Δ6MWD.

To assess echocardiographic predictors of functional improvement, an additional multiple regression model including four variables, adjusted for baseline 6MWD, age and MR grade, was calculated (see S1 Table). The model significantly predicted Δ6MWD from baseline to the four weeks follow up with moderate goodness-of-fit (p = 0.028, $R^2$ = 0.226). Apart from base-line 6MWD (B = -0.3, p = 0.001), no independent predictor was identified. Neither LVEF,

**Table 2. Echocardiographic data, functional capacity and medication at baseline and four weeks after MitraClip implantation (MCI).**

| | baseline | 4 weeks after MCI | p |
|---|---|---|---|
| NYHA functional class | | | |
| NYHA I | 0 | 11 (13.9) | **<0.001** |
| NYHA II | 2 (2.5) | 38 (48.1) | |
| NYHA III | 61 (77.2) | 15 (19.0) | |
| NYHA IV | 14 (17.7) | 0 | |
| 6MWD, m | 265 ± 103 | 295 ± 104 | **<0.001** |
| Δ6MWD, m | | 30 ± 68 | |
| MR Grade | | | |
| 1 | 0 | 56 (70.9) | **<0.001** |
| 2 | 0 | 19 (24.1) | |
| 3 | 21 (26.6) | 4 (5.1) | |
| 4 | 58 (73.4) | 0 | |
| MR PISA EROA, cm$^2$ | 0.33 ± 0.17 | 0.12 ± 0.08 | **<0.001** |
| MR PISA RVol, ml | 53 ± 27 | 18 ± 10 | **<0.001** |
| MV mean pressure gradient, mmHg | 2.4 ± 1.2 | 3.7 ± 1.8 | **<0.001** |
| Device success | | 75 (94.9) | |
| LVEF, % | 45 ± 14 | 45 ± 13 | 0.719 |
| LVEDD, mm | 59 ± 9 | 57 ± 9 | **0.035** |
| LVESD, mm | 47 ± 11 | 46 ± 11 | **0.049** |
| LAVI, ml/m$^2$ | 71 ± 36 | 71 ± 33 | 0.914 |
| sPAP, mmHg | 38 ± 12 | 36 ± 11 | 0.186 |
| Severe tricuspid regurgitation | 15 (19.0) | 19 (24.1) | 0.317 |
| ACE inhibitor | 27 (50.9) | 31 (50.0) | 1.000 |
| AT$_1$ antagonist | 13 (24.5) | 14 (22.6) | 1.000 |
| ARNI | 4 (7.5) | 5 (8.1) | 1.000 |
| β-adrenergic antagonist | 49 (92.5) | 52 (83.9) | 0.219 |
| Aldosterone antagonist | 27 (50.9) | 39 (62.9) | 0.289 |
| Loop diuretic | 49 (94.2) | 59 (95.2) | 1.000 |
| Loop diuretic dose, mg furosemide equivalent[a] | 20 [10–40] | 20 [10-60] | 0.103 |

Continuous variables are expressed as mean ± SD, continuous variables with skewed distribution as median [IQR]. Categorical variables are expressed as n (%). P-values represent results of the comparison of baseline and four weeks after MitraClip implantation using paired t-tests for continuous variables, Wilcoxon signed-rank tests for categorical variables and McNemar's test for binary variables.

[a]10 mg torasemide was converted to 20 mg furosemide equivalent.

ACE, Angiotensin-converting-enzyme; ARNI, Angiotensin receptor neprilysin inhibitor; AT$_1$, Angiotensin II receptor type 1; EROA, effective regurgitation orifice area; IQR, interquartile range; LAVI, left atrial volume index; LVEDD, left ventricular end diastolic diameter; LVEF, left ventricular ejection fraction; LVESD, left ventricular end systolic diameter; MCI, MitraClip implantation; MR, mitral regurgitation; MV, mitral valve; NYHA, New York Heart Association; PISA, proximal isovelocity surface area; RVol, regurgitation volume; SD, standard deviation; sPAP, systolic pulmonary artery pressure; 6MWD, six-minute walk distance.

LVEDD, left atrial volume index or left ventricular mass index independently predicted Δ6MWD.

## Gender-specific differences in baseline characteristics and outcomes

Considering the findings of the regression model, baseline characteristics and outcomes of the study population were analyzed for gender-specific differences. Results on baseline characteristics are shown in Table 4. Men and women were of similar age. Distribution of dilated cardiomyopathy and coronary artery disease did no differ significantly, yet there was a trend

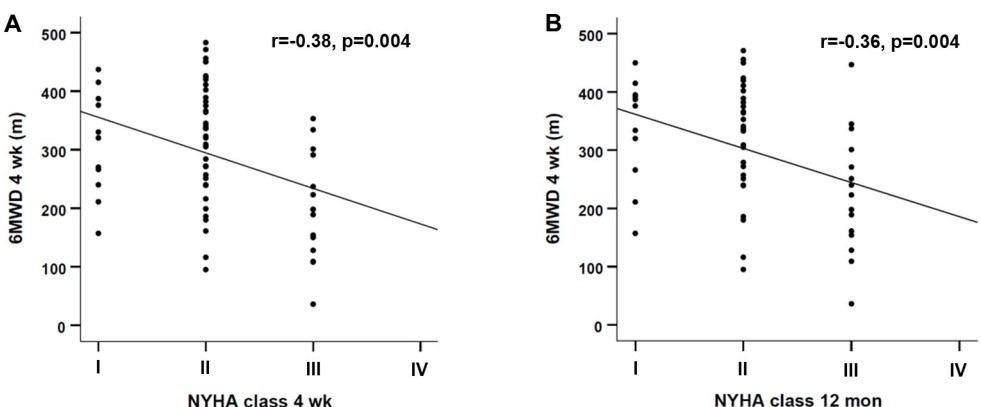

**Fig 2. Correlation between 6MWD four weeks after the procedure with NYHA functional class.** Data is shown as a scatterplot with a line of best fit. (A) Correlation between 6MWD four weeks after MCI and NYHA class four weeks after MCI. (B) Correlation between 6MWD four weeks after MCI and NYHA class twelve months after MCI. Results are expressed as Spearman's rank correlation coefficient r. MCI, MitraClip implantation; mon, months; NYHA, New York Heart Association; wk, weeks; 6MWD, six-minute walk distance.

towards a higher percentage of men suffering from coronary artery disease (69.0% vs. 48.6%, p = 0.065). Rates of atrial fibrillation, diabetes mellitus and arterial hypertension were comparable between female and male patients. Also, perioperative risk as expressed by logistic EUROScore and EUROScore II was without significant gender-related differences.

**Table 3. Clinical predictors of Δ6MWD four weeks after MitraClip implantation in the multiple linear regression model.**

|  | B | SE | β | t | p |
|---|---|---|---|---|---|
| 6MWD at baseline | -0.2 | 0.1 | -0.28 | -2.06 | **0.044** |
| Age | 0.5 | 1.2 | 0.06 | 0.41 | 0.684 |
| MR grade | 10.0 | 16.1 | 0.07 | 0.62 | 0.534 |
| LVEDD | 0.4 | 1.0 | 0.07 | 0.45 | 0.658 |
| Atrial fibrillation | -12.6 | 16.6 | -0.10 | -0.76 | 0.453 |
| Coronary artery disease | -12.7 | 16.5 | -0.10 | -0.77 | 0.446 |
| Dilated cardiomyopathy | 20.8 | 22.4 | 0.13 | 0.93 | 0.356 |
| Arterial hypertension | -39.2 | 14.6 | -0.31 | -2.67 | **0.010** |
| Diabetes mellitus | -46.9 | 14.6 | -0.37 | -3.20 | **0.002** |
| Female gender | -32.3 | 15.6 | -0.27 | -2.08 | **0.042** |
| Preserved LVEF | 14.2 | 16.0 | 0.12 | 0.88 | 0.381 |
| NYHA functional class at baseline | -4.4 | 16.4 | -0.03 | -0.27 | 0.790 |
| R | 0.622 | | | | |
| $R^2$ | 0.387 | | | | |
| Adjusted $R^2$ | 0.260 | | | | |
| F | (12, 58) = 3.048 | | | | |
| p | 0.002 | | | | |
| n | 77 | | | | |

Results on the predictors are reported as coefficient B, standard error SE, standardized coefficient β, t-statistic t and p-value. Overall model characteristics are reported as multiple correlation coefficient R, coefficient of determination $R^2$ and F-ratio F.

LVEDD, left ventricular end diastolic diameter; LVEF, left ventricular ejection fraction; MR, mitral regurgitation; NYHA, New York Heart Association; SE, standard error; 6MWD, six-minute walk distance.

**Table 4. Gender-specific differences in baseline characteristics.**

|  | male | female | p |
|---|---|---|---|
| Age, y | 76 ± 6 | 76 ± 8 | 0.974 |
| Heart failure entity |  |  |  |
| HFpEF | 12 (28.6) | 20 (54.1) | **0.032** |
| HFmrEF | 10 (23.8) | 9 (24.3) |  |
| HFrEF | 20 (47.6) | 8 (21.6) |  |
| Coronary artery disease | 29 (69.0) | 18 (48.6) | 0.065 |
| Dilated cardiomyopathy | 8 (19.0) | 4 (10.8) | 0.309 |
| Atrial fibrillation | 30 (71.4) | 21 (56.8) | 0.174 |
| Arterial hypertension | 29 (69.0) | 23 (62.2) | 0.520 |
| Diabetes mellitus | 15 (35.7) | 10 (27.0) | 0.407 |
| Chronic kidney disease | 31 (73.8) | 20 (25.3) | 0.067 |
| Logistic EuroSCORE, % | 20.1 ± 11.6 | 18.9 ± 13.2 | 0.680 |
| EuroSCORE II, % | 8.6 ± 6.7 | 6.5 ± 6.1 | 0.156 |
| NTproBNP, pg/ml | 3244 [2168-5843] | 3911 [1362-6228] | 0.923 |
| NYHA functional class |  |  |  |
| NYHA I | 0 | 0 | 0.224 |
| NYHA II | 2 (4.9) | 0 |  |
| NYHA III | 33 (80.5) | 28 (77.8) |  |
| NYHA IV | 6 (14.6) | 8 (22.2) |  |
| 6MWD, m | 288 ± 89 | 240 ± 112 | **0.034** |
| MR Grade |  |  |  |
| 1 | 0 | 0 | **0.035** |
| 2 | 0 | 0 |  |
| 3 | 7 (16.7) | 14 (37.8) |  |
| 4 | 35 (83.3) | 23 (62.2) |  |
| LVEF, % | 42 ± 14 | 49 ± 12 | **0.019** |
| LVEDD, mm | 63 ± 9 | 54 ± 7 | **< 0.001** |
| LVESD, mm | 51 ± 11 | 42 ± 10 | **< 0.001** |
| LAVI, ml/m$^2$ | 77 ± 43 | 64 ± 25 | 0.152 |

Continuous variables with normal distribution are expressed as mean ± SD, continuous variables with skewed distribution as median [IQR]. Categorical variables are expressed as n (%). P-values represent results of unpaired t-tests for continuous variables, Pearson's chi-squared tests for nominal variables and Mann–Whitney U tests for ordinal variables.

HFmrEF, heart failure with mid-range ejection fraction; HFpEF, heart failure with preserved ejection fraction; HFrEF, heart failure with reduced ejection fraction; IQR, interquartile range; LAVI, left atrial volume index; LVEDD, left ventricular end diastolic diameter; LVEF, left ventricular ejection fraction; LVESD, left ventricular end systolic diameter; MR, mitral regurgitation; MV, mitral valve; NYHA, New York Heart Association; SD, standard deviation; 6MWD, six-minute walk distance.

Gender-specific differences in echocardiographic and functional outcome four weeks after MitraClip implantation are shown in Table 5. While NYHA functional class was without gender-specific disparities, 6MWD was notably lower in women both at baseline (240 ± 112 m vs. 288 ± 89 m, p = 0.034) and four weeks after MitraClip implantation (267 ± 109 m vs. 320 ± 94 m, p = 0.024). Baseline LVEF was markedly better in female than in male patients (49 ± 12% vs. 42 ± 14%, p = 0.019), mainly driven by a higher percentage of women with preserved LVEF (54.1% of female vs 28.6% of male patients, p=0.032). Concomitantly, baseline LVEDD and LVESD were significantly smaller in female patients (54 ± 7 mm vs. 63 ± 9 mm, p < 0.001 and 42 ± 10 mm vs. 51 ± 11 mm, p < 0.001). Additionally, baseline MR grade was less severe in women than in men (grade 3 37.8% vs. 16.7%, grade 4 62.2% vs. 83.3%, p = 0.035), while

**Table 5. Gender-specific differences in echocardiographic parameters, functional outcome and medication four weeks after MitraClip implantation.**

| | male | female | p |
|---|---|---|---|
| NYHA functional class | | | |
| NYHA I | 6 (17.6) | 5 (16.7) | 0.391 |
| NYHA II | 22 (64.7) | 16 (53.3) | |
| NYHA III | 6 (17.6) | 9 (30.0) | |
| NYHA IV | 0 | 0 | |
| 6MWD, m | 320 ± 94 | 267 ± 109 | **0.024** |
| MR Grade | | | |
| 1 | 32 (76.2) | 24 (64.9) | 0.181 |
| 2 | 10 (23.8) | 9 (24.3) | |
| 3 | 0 | 4 (10.8) | |
| 4 | 0 | 0 | |
| MV mean pressure gradient, mmHg | 3.2 ± 1.2 | 4.2 ± 2.1 | **0.017** |
| LVEF, % | 41 ± 14 | 49 ± 11 | **0.008** |
| LVEDD, mm | 62 ± 9 | 53 ± 7 | $< $ **0.001** |
| LVESD, mm | 51 ± 11 | 40 ± 9 | $< $ **0.001** |
| LAVI, ml/m$^2$ | 74 ± 39 | 67 ± 23 | 0.316 |
| ACE inhibitor | 17 (53.1) | 13 (43.3) | 0.459 |
| AT$_1$ antagonist | 6 (18.8) | 8 (26.7) | 0.550 |
| ARNI | 4 (12.5) | 1 (3.3) | 0.355 |
| β-adrenergic antagonist | 29 (90.6) | 24 (80.0) | 0.294 |
| Aldosterone antagonist | 22 (68.8) | 16 (53.3) | 0.298 |
| Loop diuretic | 30 (93.8) | 28 (96.6) | 1.000 |
| Loop diuretic dose, mg furosemide equivalent[a] | 20 [10–40] | 20 [10-60] | 0.862 |

Continuous variables with normal distribution are expressed as mean ± SD, continuous variables with skewed distribution as median [IQR]. Categorical variables are expressed as n (%). P-values represent results of the comparison between male and female patients using unpaired t-tests for continuous variables, Mann–Whitney U tests for categorical variables and Fisher's exact test for binary variables.

ACE, Angiotensin-converting-enzyme; ARNI, Angiotensin receptor neprilysin inhibitor; AT$_1$, Angiotensin II receptor type 1; IQR, interquartile range; LAVI, left atrial volume index; LVEDD, left ventricular end diastolic diameter; LVEF, left ventricular ejection fraction; LVESD, left ventricular end systolic diameter; MR, mitral regurgitation; MV, mitral valve; NYHA, New York Heart Association; SD, standard deviation; 6MWD, six-minute walk distance.

[a]10 mg torasemide was converted to 20 mg furosemide equivalent.

residual MR after MitraClip therapy did not differ. However, mitral valve mean pressure gradient after MitraClip procedure was significantly higher in women (4.2 ± 2.1 mmHg vs 3.2 ± 1.2 mmHg, p = 0.017). Considering this finding, additional regression analysis adjusted for age, baseline 6MWD and MR Grade was conducted and revealed a negative correlation between post-procedural mitral valve mean pressure and Δ6MWD in the overall collective (B = -9.2, p = 0.023, see S2 Table). Heart failure medication and diuretics intake four weeks after the intervention was without gender-specific difference.

## Discussion

To the best of our knowledge, this is the first study to investigate clinical predictors of improvement in 6MWD after MitraClip implantation. Our main findings were:

1. Patients showed a significant increase in 6MWD and a decrease in NYHA class four weeks after MitraClip implantation, reflecting relevant functional improvement early after intervention. Higher postprocedural 6MWD was associated with lower NYHA class twelve months after the intervention.

2. Diabetes mellitus and arterial hypertension were negative predictors of improvement in 6MWD after MitraClip implantation.

3. Female gender was a negative predictor of increase in 6MWD after MitraClip therapy, with women presenting more often with preserved LVEF, less LV dilatation and higher postprocedural mitral valve mean pressure gradient than male patients.

Identifying the patients who profit from transcatheter mitral valve therapy is crucial for both achieving therapeutic success and avoiding futile interventions [13]. As heart failure symptoms like exertional dyspnea are a pivotal criterion for patient selection in current guidelines [23], symptom relief is a main therapeutic goal of transcatheter mitral valve repair. Therefore, improvement in functional capacity as expressed by an increase in 6MWD is an important aspect of the outcome of MitraClip therapy. Our study identified several factors indicative of worse functional outcome which could aid to improve patient selection and generate new hypotheses on the determinants of therapeutic response to transcatheter mitral valve repair.

## Gender-related differences in functional outcome of MitraClip therapy

Our analysis identified female gender to be a negative predictor of improvement in 6MWD after MitraClip implantation, expressing a less favourable functional outcome in women compared to men. Significant gender-specific differences have been reported in the outcome after mitral valve surgery, with female patients exhibiting both higher short- and long-term mortality [24–27]. Also, surgical mitral valve repair restored life expectancy to normal compared to matched controls in men, but not in women [25]. According to previous investigations, this does not apply to transcatheter mitral valve repair. In several registries, short- and long-term mortality was equal between men and women, with one study even reporting superior long-term survival in women [28–32]. However, congruous with our observations, gender-related differences were noted in the functional improvement after transcatheter mitral valve repair. In the TRAMI registry, female patients exhibited less improvement in functional NYHA class one year after MitraClip implantation than male patients [30]. In another retrospective study, Tigges et al reported an increase in 6MWD only in men, while it stagnated in women [29]. Notably, procedural efficacy in reducing MR did not differ between males and females. Furthermore, a subgroup analysis of the randomized COAPT-Trial also supports our observations. Patients' gender nearly significantly interacted with the rate of hospitalization for heart failure, with a notable trend towards worse outcome in women [10].

The reasons for the observed gender-related differences in functional outcome are not clear and most likely multifactorial. In our study population, women and men showed comparable age, perioperative risk and non-cardiac comorbidities. However, female patients presented more commonly with preserved LVEF and less LV dilatation than male patients. Given similar findings in other registry studies [28–30], these gender-specific disparities in baseline characteristics appear to be inherent in the population treated with transcatheter mitral valve repair. Worse functional outcome in women might implicate that therapeutic response to MitraClip implantation is less effective in patients with preserved LVEF. Conversely, differences in LV geometry between male and female patients may constitute a major cause for female gender negatively predicting functional improvement. Furthermore, smaller and different mitral valve

morphologies in women [26], which are mirrored by the lower number of clips implanted in female patients [29,30], possibly pose a higher challenge to transcatheter mitral valve repair. This might particularly lead to higher postprocedural mitral valve pressure gradients in women than in men, as it was observed in our study. Importantly in this context, postprocedural mitral valve pressure gradient negatively correlated with improvement in 6MWD in the overall study population. Elevated mitral valve pressure gradient might increase left atrial pressure, of which the latter is associated with less improvement in 6MWD after MitraClip implantation [33]. A recent trial on MitraClip therapy for secondary MR did not show a correlation between postprocedural mitral valve pressure gradient and 6MWD, but with worse NYHA class [34]. Therefore, elevated mitral valve pressure gradient in women might be an important factor in the observed gender-specific difference.

In our study population, women tended to have less severe MR at baseline, which could lead to the assumption that our observations are not genuinely gender-related. However, functional outcome was equal between patients with MR grade 3 and grade 4. Besides, given the lack of indexed cut-off parameters in MR grading [22], echocardiography might underestimate MR severity in female patients. This could also lead to a delay in diagnosis and treatment in a later stage of the disease.

While female gender was identified as a negative predictor for improvement in 6MWD, NYHA functional class was not significantly different between male and female patients. This is most likely due to the fact that NYHA functional class is a highly subjective and approximate measure of functional status, relying on the physician's opinion derived from patient's history [35]. Indeed, clinical research demonstrated that reproducibility of NYHA classification when assessed by two independent physicians is only 56% [36]. The 6MWD, however, is an objective measure of functional performance with good reproducibility in patients with heart failure [37,38]. Hence, in our study, NYHA classification might not have been sensitive enough to detect the gender-specific difference in functional outcome which was observed in the results of the 6MWD. Previous research reported an increase in 6MWD after repeated administration within one day or one week in the absence of any intervention, demonstrating a learning or training effect [39,40]. However, as our study population only performed 6MWD twice and four weeks apart and considering the immobilization during the hospital stay, a significant training effect seems unlikely. Still, it cannot be ruled out when interpreting the results.

Apart from reasons of physical nature, socioeconomic factors and health behavior might also contribute to the observed gender-specific differences. As our study population was treated at a center for transcatheter valve repair, evaluation for MitraClip therapy relied on the referral of symptomatic patients by general practitioners and external cardiologists. It is possible that women underexaggerate their disease or display atypical symptoms, leading to delayed treatment. Concomitantly, experience from mitral valve surgery shows that women are less likely to receive elective therapy, but present more frequently on an urgent basis and with advanced disease [27]. Thus, in the context of our findings, valve repair might be performed at a stage when MR already inflicted irreversible impairment of ventricular and atrial function. On the other hand, it is also thinkable that women are more sensitive in perceiving symptoms of MR such as dyspnea. In previous studies on the clinical care of heart failure, women experienced more symptoms than men [41,42]. This might result in a larger proportion of female patients with clinical less significant MR and worse functional improvement after MitraClip procedure, consecutively. Furthermore, patients' gender might also influence perception and decision of medical personnel and family members when considering performing an invasive therapy in this elderly patient population.

## Impact of diabetes mellitus und arterial hypertension on the outcome of MitraClip implantation

Diabetes mellitus and arterial hypertension are very frequent comorbidities in patients with secondary MR [4]. In this study, both diseases were identified as negative predictors of the functional improvement after MitraClip procedure. Consistent with our results, diabetes mellitus was found to be a determinant of NT-proBNP nonresponse after MitraClip implantation [43]. In a large MitraClip registry study, it was also an independent predictor of 1-year mortality [44]. Diabetes is known to cause diabetic cardiomyopathy, which is characterized by myocardial interstitial fibrosis and extracellular remodeling, leading to LV hypertrophy and reduced LV compliance [45]. Arterial hypertension also classically induces LV hypertrophy with myocardial fibrosis [46,47]. Thus, both diseases contribute to the development of LV diastolic dysfunction with increased myocardial stiffness and elevated LV filling pressures.

Our results might indicate that these hemodynamic changes respond less favorable to a reduction of secondary MR by transcatheter repair. Furthermore, differences in the pathophysiology of MR induced by diastolic dysfunction as opposed to MR caused by severe systolic dysfunction could affect the success of MitraClip therapy. Interestingly, these considerations fit to our observations concerning gender-related differences as described above, as female patients more often had preserved LVEF and less LV dilation. Apart from left atrial volume, whose validity in estimating LV filling pressure is limited in the presence of MR [48], our study did not investigate additional parameters of diastolic function. Thus, this interpretation remains hypothetical and should be investigated in future studies.

Previous randomized controlled studies on the efficacy of transcatheter mitral valve repair in secondary MR only included patients with severe LV systolic dysfunction [9,10]. In contrast, our study and other large registries report that in common practice a significant proportion of patients treated with MitraClip implantation for secondary MR has preserved LVEF [8,30]. This calls for further investigation of the efficacy of transcatheter mitral valve repair in patients with secondary MR and heart failure with preserved ejection fraction.

## Limitations

Our study might have some limitations. It presents outcome data of an experienced academic institution and might therefore not be transferable to other MitraClip centers. Participants without four-weeks follow-up and/or a missing 6MWD at baseline were excluded from the analysis. This markedly reduced the study population, thus potentially limiting the statistical power of our analysis. As the reasons for not conducting 6MWD or the follow-up were not reported, attrition bias cannot be ruled out. Hence, our results need to be considered exploratory and should be validated in a greater cohort. Still, our sample size is comparable to other registry studies on the outcome of MitraClip therapy [49,50]. As data on 6MWD at later timepoints was not available, our findings might not be applicable to long-term functional outcome in full extent. However, NYHA class remained stable after twelve months and strongly correlated with short-term 6MWD. Due to the retrospective analysis and lack of a control group, clinical and echocardiographic assessment might have been influenced by observation bias. However, the six-minute walk test is considered to be robust against bias and is widely used in the assessment of patients with cardiopulmonary diseases because of its validity and reliability [19].

Although functional capacity is an important aspect in the treatment of patients suffering from heart failure, we acknowledge that the decision for MitraClip therapy should primarily be based on mortality and morbidity outcomes. Therefore, guidance for patient selection cannot be given solely on the basis of our results. Unlike several previous registry studies on

predictors of outcome [16,18,49], we only included patients with MR of secondary origin, given its distinct pathophysiologic difference to primary MR. Nevertheless, the mechanisms causing secondary MR and the underlying heart disease are manifold [12]. Consequently, our study collective remains considerably heterogenous, reflecting the patient population treated with transcatheter mitral valve repair. Hence, the identified predictors might not be applicable to all subgroups of the collective. Further subgroup and sensitivity analysis, which could not be conducted in this study due to the sample size, would be useful to confirm our results. Last, we acknowledge that a complete evaluation of functional outcome after MitraClip therapy should also incorporate information on quality of life, which could not be collected in our study due to its retrospective design.

## Conclusions

In this retrospective study, we confirmed that MitraClip therapy of secondary MR in a real-world and high-risk collective improved functional capacity even in the short-term. Female gender was found to be a negative predictor of functional improvement, with women exhibiting preserved LVEF more often, less LV dilatation and higher postprocedural mitral valve pressure gradient. Diabetes mellitus and arterial hypertension, both diseases associated with LV hypertrophy and diastolic dysfunction, were also found to be predictors of less favourable functional outcome. While further validation of the results in a larger cohort is recommended, these parameters may be used to improve patient selection for MitraClip therapy.

## Supporting information

**S1 Table. Echocardiographic predictors of Δ6MWD four weeks after MitraClip implantation.**
(DOCX)

**S2 Table. Correlation between post-procedural mitral valve mean pressure gradient and Δ6MWD.**
(DOCX)

**S1 Dataset. Complete raw data of the study.**
(XLSX)

## Author Contributions

**Conceptualization:** Bernhard Unsöld, Christoph Birner.

**Data curation:** Michael G. Paulus, Lukas Böhm, Magdalena Holzapfel.

**Formal analysis:** Michael G. Paulus, Lukas Böhm, Magdalena Holzapfel.

**Investigation:** Michael G. Paulus, Christine Meindl, Michael Hamerle, Christian Schach, Kurt Debl.

**Methodology:** Michael G. Paulus, Christine Meindl.

**Project administration:** Christine Meindl, Bernhard Unsöld, Christoph Birner.

**Resources:** Lars S. Maier, Kurt Debl, Bernhard Unsöld, Christoph Birner.

**Supervision:** Lars S. Maier, Bernhard Unsöld, Christoph Birner.

**Writing – original draft:** Michael G. Paulus.

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
