## [Decision Letter · Decision Letter 0]

9 Dec 2019

PONE-D-19-24140

Predictors of functional improvement in the short term after MitraClip implantation in patients with secondary mitral regurgitation

PLOS ONE

Dear Dr. Paulus,

Thank you for submitting your manuscript to PLOS ONE. After careful consideration, we feel that it has merit but does not fully meet PLOS ONE’s publication criteria as it currently stands. Therefore, we invite you to submit a revised version of the manuscript that addresses the points raised during the review process and quoted below.

We would appreciate receiving your revised manuscript by Jan 23 2020 11:59PM. To enhance the reproducibility of your results, we recommend that if applicable you deposit your laboratory protocols in protocols.io, where a protocol can be assigned its own identifier (DOI) such that it can be cited independently in the future. For instructions see: http://journals.plos.org/plosone/s/submission-guidelines#loc-laboratory-protocols

We look forward to receiving your revised manuscript.

Kind regards,

Marc W. Merx, MD

Academic Editor

PLOS ONE

Journal Requirements:

Reviewers' comments:

Reviewer's Responses to Questions

**Comments to the Author**

1. Is the manuscript technically sound, and do the data support the conclusions?

Reviewer #1: Yes

Reviewer #2: Yes

2. Has the statistical analysis been performed appropriately and rigorously? 

Reviewer #1: Yes

Reviewer #2: No

3. Have the authors made all data underlying the findings in their manuscript fully available?

Reviewer #1: Yes

Reviewer #2: No

4. Is the manuscript presented in an intelligible fashion and written in standard English?

Reviewer #1: Yes

Reviewer #2: No

5. Review Comments to the Author

Reviewer #1: In the present paper from Paulus et al. the authors retrospectively analyzed 79 patients with secondary MR treated with MitraClip implantation. Before and four weeks after the procedure, the patients underwent 6MWD and echocardiography. In the general population 6MWD significantly improved. However, female gender, diabetes mellitus and arterial hypertension were found to be predictors of reduced 6MWD improvement. The authors concluded that MitraClip therapy improves functional capacity in high-risk patients even in the short term of four weeks after the procedure. Female gender, diabetes mellitus and arterial hypertension are baseline predictors of a less favorable functional outcome. At least in the reviewers opinition, the following comments have to be addressed:

Major points:

1. The main query relates to the clinical impact of the study and the conclusion of the authors. Selecting the endpoint “improvement in 6-MWT four weeks following PMVR” simply cannot serve as any guidance for patient selection. In the current age, the majority of patients who undergo PMVR do have hypertension, every second patient is female and about every fifth patient suffers from diabetes. Patient selection for PMVR should be performed on the basis of “hard endpoint” trials, thus the clinical benefit of the current trial appears somewhat limited.

2. Trial inclusion went on over a total of nine years. Please indicate the reason for the rather small number of participants. Is there a selection bias compared to the all-comers clip cohort in your center? Even more important, please indicate how outcome in terms of 6MWT 4 weeks following PMVR correspond to NYHA class or survival at later timepoints, which should be available in the majority of participants.

3. Data describing the interaction between post-procedural MVPG and improvement in 6MWT are missing. This could be of prime importance, because the authors stated that diastolic dysfunction (hypertension and diabetes) might be the reason for the observation that men seem to benefit more than women. Gradients created by PMVR will at further resistance to the already impaired diastolic filling. As a matter of fact, in the current trial women tended to have higher post-procedural MVPG than men (p=0.017). In theory, this may explain the less improvement in 6MWT in women. Previous data investigated the impact of post-procedural MVPG and LA pressure and found that a lower MVPG and reduction in LA pressure were predictors of improvement in 6MWT. Therefore, data comparing the post-procedureal MVPG in patients with and without improvement in 6MWT need to be addressed.

Minor points:

4. The authors describe that they “solely included patients with MR of secondary origin, given its distinct pathophysiologic difference to primary MR”. As a matter of fact, even secondary MR comprises of numerous entities (e.g. ischemic versus dilative, thethering of the leaflets, atrial functional MR), who at least in part show distinct survival and distinct response to PMVR by MitraClip. In other words, the small cohort referred to in the manuscript may not be as homogenous at it appears on first glance. This should be clarified.

5. The fact that 6MWT following 4 weeks after MitraClip is higher in men than in women albeit the heart failure symptoms according to NYHA class are identical needs further clarification. Could it be possible that man react different to repetitive 6MWT than women in terms of training effects? Relative increases in repetitive 6MWD in the range of 5% have been shown in the literature.

6. Another important fact is the change of medication, e.g. in diuretic therapy. In many cases, medication is changed after MitraClip implantation with higher doses of diuretics which might also influence functional capacity and 6MWT. Therefore, it would be helpful if the authors provided data on medication prior and after MitraClip implantation.

7. In results section and tables NT-proBNP is presented as median and standard deviation which should be median and percentiles.

8. The authors have to be congratulated for investigating functional outcome following PMVR. However, the manuscript in its current form applies 6MWT, only which is certainly one piece in the puzzle of quality of life only. It would be extremely helpful if the 6MWT data could be underscored by corroborating quality of life questionnaires.

Reviewer #2: General comments:

The authors aimed to elucidate predictors of symptomatic improvement by means of 6 minutes walk distance (6MWD) after MitraClip for the treatment of functional severe MR. Secondary MR is a very common valvular heart disease and associated with poor prognosis in patients with heart failure. While the two recent RCTs (Mitra-FR and COAPT) showed the competing effect of MitraClip on all-cause mortality and HF rehospitalization, however, one of the main goal of this therapy is also to improve functional capacity. The topic could be interesting and the findings are hypothesis generating. However, several aspects should be nonetheless addressed.

My comments are below:

・Statistical analysis) “a multiple linear regression model was calculated to identify independent predictors of improvement in 6MWD after MitraClip implantation. Selection of predictors was based on clinical considerations and exploratory statistical analysis.” Please consider also age and baseline echocardiographic parameters (e.g. LVEDD, severity of MR) as covariates, because these parameters are supposed to be associated with functional capacity.

・Results) The authors determined the history of hypertension, diabetes, and female sex are negative indicators of symptomatic improvement by means of 6MWD. Please consider to perform sensitivity analyses to confirm these results. For example, are the associations of these factors consistent in the subgroup (e.g., age ≤75 or >75, male or female sex etc)？

・Results) Female sex, hypertension, and diabetes mellitus might suggest that HEFPEF or diastolic function could be associated with poor improvement of symptomatic status after MitraClip. Please consider to conduct an additional model including echo parameters, such as LVEF, LAVI, and LVMI, which are indicating HEFPEF and diastolic function.

・Discussion) “Performing MitraClip implantation in secondary MR in a high-risk collective resulted in a high device success rate even in the short-term”. But the authors are not able to mention this, because the authors excluded patients who failed to implant MitraClip and those who did not complete follow-up examinations. Please modify or omit this sentence.

・Limitations) The study included 79 patients who underwent MitraClip and were available for data of 6MWD. The limited sample size and the study population render the results and conclusion dubious. Because the present study was based on the population only who were able to perform FU-6 MWD, so I’m not sure that the indicated clinical predictors are useful to improve patient selection, in our clinical practice. How many consecutive patients were treated with MitraClip in the study period？ And how many patients were excluded due to No technical success or No FU-assessment？

Minor comments:

・ Please define MR grade in the methods section.

・ English proofreading may be required.

6. PLOS authors have the option to publish the peer review history of their article (what does this mean?). If published, this will include your full peer review and any attached files.

Reviewer #1: No

Reviewer #2: No

---

## [Author Response · Author response to Decision Letter 0]

15 Jan 2020

REVIEWER 1

We gratefully appreciate that Reviewer 1 acknowledges our investigation of the functional outcome after MitraClip implantation. We are also very thankful for the valuable comments which we would like to address as follows.

The Reviewer expresses that the trial’s clinical impact and its use for patient selection is limited because it did not investigate hard endpoints.

We clearly agree with the Reviewer’s opinion that patient selection for MitraClip implantation should primarily be based on hard endpoints like mortality. Nevertheless, in our clinical experience, patients who consider undergoing MitraClip expect symptom relief and improvement in their functional status, which is why clinical endpoints are increasingly acknowledged as relevant outcome parameters. Therefore, in our opinion, short-term improvement in functional capacity as expressed by an increase in 6MWT is still an important aspect of the outcome of MitraClip therapy. This is especially true for the collective of patients with secondary mitral regurgitation, whose prognosis is often significantly limited by the severe comorbidities. Yet, we acknowledge that while our study adds new hypothesis-generating aspects, guidance for patient selection cannot be given solely on the basis of our results. In our revised manuscript, this is clarified in the discussion section.

The Reviewer asks for details on the inclusion process and possible selection bias, as he perceives a discrepancy between the number of included patients and the length of the inclusion period.

To more clearly demonstrate the inclusion process of our study, we added a detailed flowchart to the methods section of our revised manuscript which explicitly states the total number of patients who underwent MitraClip therapy at our center during the inclusion period. The most frequent reason for exclusion was MR of primary origin. As in many other MitraClip centres, the quantity of procedures increased over the years, clearly reflecting the growing clinical acceptance of this therapeutic modality. Due to the retrospective nature of our study, we do not have information on the reasons for patients not completing the follow-up examinations. Consequently, attrition bias cannot be ruled out when interpreting our results. Therefore, in the revised manuscript, we added this fact as a limitation in the discussion section.

The Reviewer asks for data on NYHA class and mortality at later timepoints and their correlation with short-term increase in 6MWD.

We agree with the Reviewer that the correlation between improvement in 6MWD and long-term functional class is an important aspect and are thankful for this valuable comment. Therefore, in the revised manuscript, we included data on NYHA functional class twelve months after the procedure which is available for the majority of patients. Additionally, further statistical analysis revealed a moderate correlation between postprocedural 6MWD and NYHA functional class both four weeks and twelve months after the procedure.

Because our study design was dedicated to functional outcome after MitraClip therapy, we did not evaluate mortality endpoints. 

The Reviewer requests an analysis of the interaction between post-procedural MVPG and improvement in 6MWD.

We are grateful for this excellent remark. In the revised manuscript, we conducted an additional analysis, revealing a negative correlation between post-procedural MVPG and increase in 6MWD. This indeed may play an important role in the observed gender-specific differences, which is outlined in the discussion section of the revised manuscript.

The Reviewer states that the trial’s cohort, although limited to patients with secondary MR, is heterogenous regarding the pathomechanism of MR.

The Reviewer is right that our study population comprises numerous entities of secondary MR and underlying structural heart disease, reflecting the patient collective treated with MitraClip implantation. In the revised manuscript, this is clarified in the discussion section.

The Reviewer requests further clarification of the aspect that while female gender predicted worse improvement in 6MWD, NYHA functional class is not different between male and female patients. Additionally, the Reviewer asks if assessment of 6MWD in our study might have been subject to a learning effect.

Indeed, while female gender was identified as an independent predictor of less improvement in 6MWD, NYHA class was not significantly different between male and female patients. This is likely explained by the limitations of the NYHA classification: It is subjective and exhibits a rather low reproducibility rate. In contrast, the 6MWT is an objective measure of functional performance with high reproducibility in patients with heart failure. Therefore, we consider the 6MWT to be superior in detecting differences in functional outcome. In the revised manuscript, this is clarified in the discussion section.

The Reviewer is right that 6MWT exhibits a training effect when conducted repeatedly during one day or one week. However, as our study population only performed 6MWT twice and four weeks apart and considering the immobilization during the hospital stay, we consider a significant training to be unlikely. Still, it cannot be completely ruled out when interpreting the results.

The Reviewer requests data on medication before and after MitraClip therapy, as this might also influence functional capacity.

We agree that a change in medication, particularly diuretics intake, might have an impact on functional capacity. Therefore, in the revised manuscript, we included data on medication at baseline and four weeks after the procedure, which did not change significantly. Also, medication four weeks after MitraClip therapy did not differ between men and women.

The reviewer suggests that the data on NTproBNP should be presented as median and percentiles.

We are thankful for this justified remark. In the revised manuscript, NTproBNP is presented as median and percentiles.

The reviewer asks for data of quality of life questionnaires.

We clearly agree with the reviewer that quality of life questionnaires are very important tools for further investigation of functional outcome. As our study was retrospective and quality of life questionnaires were not part of our routine clinical care, we cannot provide these additional data. In our revised manuscript, this aspect was added as a limitation in the discussion section.

REVIEWER 2

We are thankful that the Reviewer finds our manuscript interesting and its findings hypothesis-generating. We also appreciate the useful comments to which we would like to respond as follows:

The Reviewer suggests adding age and baseline echocardiographic parameters such as MR grade as covariates in the linear regression model.

We appreciate this valuable recommendation. In the revised manuscript, age and baseline MR grade were added as covariates in the linear regression model.

The Reviewer recommends performing sensitivity or subgroup analyses.

We acknowledge that conducting sensitivity analysis would be useful to further confirm the validity of the identified predictors. However, this possibility is restricted by the sample size. Performing the regression model on a subgroup (e.g. female patients) would reduce the sample size to such an extent that overfitting of the model occurs, leading to invalid results. We added this aspect in the limitations section of the revised manuscript. 

The Reviewer asks for an additional regression model which includes echocardiographic parameters, particularly parameters of diastolic dysfunction.

We are grateful for this recommendation. In the revised manuscript, we conducted an additional regression model which includes several echocardiographic parameters. No independent echocardiographic predictor was identified. As a measure for diastolic dysfunction, left ventricular mass index and left atrial volume index were analyzed. As tissue doppler imaging was not part of our clinical routine follow up, we are not able to provide more specific parameters of diastolic function.

The Reviewer states that our discussion cannot draw conclusions on device success rate as patients who failed to implant MitraClip or did not complete follow-up were omitted from analysis.

We fully agree that our study is not able to draw conclusions on device success due to its study design. Therefore, we omitted this sentence in the revised manuscript.

The Reviewer expresses that the impact of the study is limited by the sample size and asks for further details on the number of excluded patients.

To elucidate the inclusion process, we included a study flow chart to the methods section of the revised manuscript, depicting the number of patients who underwent MitraClip implantation at our center during the inclusion period as well as the number of patients who were excluded from analysis. Exclusion because of failure to implant a clip was rare, occurring only in six patients. Patients without a 6MWT were excluded in the intent of a complete case analysis. As we do not have information on the reasons why patient did not perform 6MWT, we agree that attrition bias cannot be ruled out when interpreting our results. In the revised manuscript, this was added as a limitation in the discussion section.

As mentioned in the discussion section of our manuscript, we agree that the sample size could be seen as a limitation to our study. While its results need to be validated in a larger cohort, our trial may aid to improve patient selection and generate new hypotheses on the mechanisms determining therapeutic response to MitraClip implantation.

The Reviewer asks for definition of MR grade in the methods section.

Corresponding to this comment, we detailed MR grading in the methods section.

---

## [Decision Letter · Decision Letter 1]

25 Mar 2020

PONE-D-19-24140R1

Predictors of functional improvement in the short term after MitraClip implantation in patients with secondary mitral regurgitation

PLOS ONE

Dear Dr. Paulus,

Thank you for submitting your manuscript to PLOS ONE. After careful consideration, we feel that it has merit but does not fully meet PLOS ONE’s publication criteria as it currently stands. Therefore, we invite you to submit a revised version of the manuscript that addresses the points raised during the review process. Please be sure to consider the additional points raised by reviewer#4 and try and add some of the aspects to the next Version of your mansucript.

We would appreciate receiving your revised manuscript by May 09 2020 11:59PM. To enhance the reproducibility of your results, we recommend that if applicable you deposit your laboratory protocols in protocols.io, where a protocol can be assigned its own identifier (DOI) such that it can be cited independently in the future. For instructions see: http://journals.plos.org/plosone/s/submission-guidelines#loc-laboratory-protocols

We look forward to receiving your revised manuscript.

Kind regards,

Marc W. Merx, MD

Academic Editor

PLOS ONE

Reviewers' comments:

Reviewer's Responses to Questions

**Comments to the Author**

1. If the authors have adequately addressed your comments raised in a previous round of review and you feel that this manuscript is now acceptable for publication, you may indicate that here to bypass the “Comments to the Author” section, enter your conflict of interest statement in the “Confidential to Editor” section, and submit your "Accept" recommendation.

Reviewer #3: All comments have been addressed

Reviewer #4: (No Response)

2. Is the manuscript technically sound, and do the data support the conclusions?

Reviewer #3: Yes

Reviewer #4: Yes

3. Has the statistical analysis been performed appropriately and rigorously? 

Reviewer #3: Yes

Reviewer #4: Yes

4. Have the authors made all data underlying the findings in their manuscript fully available?

Reviewer #3: Yes

Reviewer #4: Yes

5. Is the manuscript presented in an intelligible fashion and written in standard English?

Reviewer #3: Yes

Reviewer #4: Yes

6. Review Comments to the Author

Reviewer #3: The topic is highly relevant and of interest. In the revised manuscript the authors have fully addressed the reviewer's concerns. I have no further comments/ concerns.

Reviewer #4: In this manuscript, Paulus et al. present retrospective data on the improvement of 6MWD 4 weeks after MitraClip procedure in 79 patients with functional mitral regurgitation. They show that 6MWD improves significantly. Their main finding is, however, that female gender is negatively associated with 6MWD improvement. Although functional outcomes are important, we doubt that the difference in improvement of 6MWD is only attributable to female gender. This is due to the many differences between the male and the female group (e.g. LV-EF, LVEDD, LVESD, NYHA class, HFpEF/HFrEF and many more). We think that the main message of the paper could be strengthened by adding of at least some of these parameters to the multivariate regression model.

Gender-related differences in 6MWD improvement have been observed 4 weeks after MitraClip procedure. Do females improve later? Are there any data on 6MWD at later timepoints?

Authors switch between the terms “6MWT” and “6MWD”. Since this is the main outcome, it should be consistent

7. PLOS authors have the option to publish the peer review history of their article (what does this mean?). If published, this will include your full peer review and any attached files.

Reviewer #3: No

Reviewer #4: No

---

## [Author Response · Author response to Decision Letter 1]

7 Apr 2020

REVIEWER 4

We thank Reviewer 4 for thoroughly reviewing our manuscript and appreciate the valuable comments which we would like to address as follows.

The Reviewer expresses that the observed gender-specific difference in functional outcome might in part be caused by differences in the baseline characteristics between male and female patients. For further clarification, the Reviewer recommends adding baseline parameters with observed gender-specific differences to the regression model.

We are grateful for this excellent remark. In the revised manuscript, we added baseline LVEDD, preserved LVEF and NYHA functional class as variables to the regression model. After correcting for these parameters, female gender remains a significant predictor of 6MWD improvement, confirming our findings. Yet, we fully agree with the Reviewer’s opinion that the reasons for the observed gender-related difference are multifactorial and may in part be explained by differences in baseline characteristics, especially in LV geometry and systolic function. However, given similar findings in the analysis of the GRASP and TRAMI registry, these gender-specific differences in baseline characteristics do not occur exclusively in our patient collective, but appear to be inherent in the population treated with TMVR. In the revised manuscript, these considerations were added to the discussion section.

The Reviewer asks for data on 6MWD improvement at later timepoints.

We agree with the Reviewer that, although short-term 6MWD strongly correlated with long-term NYHA class in our cohort, data on 6MWD at later timepoints could give further insight into determinants of functional outcome. However, as evaluation of long-term 6MWD was not part of our routine follow-up, we do not have sufficient data for valid statistical analysis. In the revised manuscript, this aspect was added as a limitation in the discussion section.

The Reviewer suggests a consistent use of either “6MWD” or “6MWT” as an abbreviation for the six-minute walk distance.

In the previous version of our manuscript, the abbreviation “6MWT” was used when referring to the walk test itself while “6MWD” referred to the result of the walk test. We clearly agree with the reviewer that this might confuse readers. Therefore, in the revised manuscript, we now consistently use the abbreviation “6MWD”.

---

## [Decision Letter · Decision Letter 2]

23 Apr 2020

Predictors of functional improvement in the short term after MitraClip implantation in patients with secondary mitral regurgitation

PONE-D-19-24140R2

Dear Dr. Paulus,

We are pleased to inform you that your manuscript has been judged scientifically suitable for publication and will be formally accepted for publication once it complies with all outstanding technical requirements.

With kind regards,

Marc W. Merx, MD

Academic Editor

PLOS ONE

Additional Editor Comments (optional):

Reviewers' comments:

Reviewer's Responses to Questions

**Comments to the Author**

1. If the authors have adequately addressed your comments raised in a previous round of review and you feel that this manuscript is now acceptable for publication, you may indicate that here to bypass the “Comments to the Author” section, enter your conflict of interest statement in the “Confidential to Editor” section, and submit your "Accept" recommendation.

Reviewer #4: All comments have been addressed

2. Is the manuscript technically sound, and do the data support the conclusions?

Reviewer #4: Yes

3. Has the statistical analysis been performed appropriately and rigorously? 

Reviewer #4: Yes

4. Have the authors made all data underlying the findings in their manuscript fully available?

Reviewer #4: Yes

5. Is the manuscript presented in an intelligible fashion and written in standard English?

Reviewer #4: Yes

6. Review Comments to the Author

Reviewer #4: (No Response)

7. PLOS authors have the option to publish the peer review history of their article (what does this mean?). If published, this will include your full peer review and any attached files.

Reviewer #4: No

---

## [Editor Report · Acceptance letter]

29 Apr 2020

PONE-D-19-24140R2 

Predictors of functional improvement in the short term after MitraClip implantation in patients with secondary mitral regurgitation 

Dear Dr. Paulus:

I am pleased to inform you that your manuscript has been deemed suitable for publication in PLOS ONE. Congratulations! Your manuscript is now with our production department. 

With kind regards,

on behalf of

Prof. Dr. Marc W. Merx 

Academic Editor

PLOS ONE